Diversity and ice nucleation activity of Pseudomonas syringae in drone-based water samples from eight lakes in Austria

http://orcid.org/0000-0003-1336-2601 Hanlon Regina 1
Jimenez-Sanchez Celia 1
Benson James 1
Aho Ken 2
Morris Cindy 3
http://orcid.org/0000-0001-5971-2502 Seifried Teresa M. 4
Baloh Philipp 4
http://orcid.org/0000-0002-2715-1429 Grothe Hinrich 4
http://orcid.org/0000-0002-7003-7429 Schmale David 1 dschmale@vt.edu
1 School of Plant and Environmental Sciences, Virginia Polytechnic Institute and State University (Virginia Tech) , Blacksburg, Virginia , United States
2 Department of Biological Sciences, Idaho State University , Pocatello, Idaho , United States
3 Institut National de la Recherche pour l’Agriculture, l’Alimentation et l’Environnement (INRAE) , Montfavet , France
4 Faculty of Technical Chemistry, TU Wien, Institute of Materials Chemistry , Vienna , Austria
Beman Michael
Electronic publication date: 2023 Nov 28
Publication date: 2023
Volume: 11
Electronic Location ID: e16390
Received 2023 Apr 26; Accepted 2023 Oct 11
Copyright: © 2023 Hanlon et al.
Copyright year: 2023
Copyright holder: Hanlon et al.
License: This is an open access article distributed under the terms of the Creative Commons Attribution License, which permits unrestricted use, distribution, reproduction and adaptation in any medium and for any purpose provided that it is properly attributed. For attribution, the original author(s), title, publication source (PeerJ) and either DOI or URL of the article must be cited.
License URL: https://creativecommons.org/licenses/by/4.0/

Keywords: Pseudomonas syringae, Ice nucleation, Diversity, Richness, Drone, Bacteria, Austria, Lake

Funding: Institute of Critical Technology and Applied Science (ICTAS) #177220 College of Agriculture and Life Sciences #137605 National Science Foundation (NSF) DEB-1241068, AGS-1520825, IIS-1637915 Austrian Science Fund (FWF) P26040 Austrian Research Promotion Agency (FFG) This research was supported by grants to David Schmale from the Institute of Critical Technology and Applied Science (ICTAS) at Virginia Tech (#177220), the College of Agriculture and Life Sciences at Virginia Tech (#137605), and the National Science Foundation (NSF) under Grant Numbers DEB-1241068 (Dimensions: Collaborative Research: Research on Airborne Ice-Nucleating Species (RAINS)), AGS-1520825 (HAZARDS SEES: Uncovering the Hidden Skeleton of Environmental Flows: Advanced Lagrangian Methods for Hazard Prediction, Mitigation, and Response), and IIS-1637915 (NRI: Coordinated Detection and Tracking of Hazardous Agents with Aerial and Aquatic Robots to Inform Emergency Responders). This research was supported by grants to Hinrich Grothe from the Austrian Science Fund (FWF) under Grant P26040, and bridge project 850689 (EarlySnow) from the Austrian Research Promotion Agency (FFG). Any opinions, findings, and conclusions or recommendations expressed in this material are those of the authors and do not necessarily reflect the views of the sponsors of this research. The funders had no role in study design, data collection and analysis, decision to publish, or preparation of the manuscript.

==============================
Bacteria from the Pseudomonas syringae complex (comprised of at least 15 recognized species and more than 60 different pathovars of P. syringae sensu stricto) have been cultured from clouds, rain, snow, streams, rivers, and lakes. Some strains of P. syringae express an ice nucleation protein (hereafter referred to as ice+) that catalyzes the heterogeneous freezing of water. Though P. syringae has been sampled intensively from freshwater sources in the U.S. and France, little is known about the genetic diversity and ice nucleation activity of P. syringae in other parts of the world. We investigated the haplotype diversity and ice nucleation activity at −8 °C (ice+) of strains of P. syringae from water samples collected with drones in eight freshwater lakes in Austria. A phylogenetic analysis of citrate synthase (cts) sequences from 271 strains of bacteria isolated from a semi-selective medium for Pseudomonas revealed that 69% (188/271) belonged to the P. syringae complex and represented 32 haplotypes in phylogroups 1, 2, 7, 9, 10, 13, 14 and 15. Strains within the P. syringae complex were identified in all eight lakes, and seven lakes contained ice+ strains. Partial 16S rDNA sequences were analyzed from a total of 492 pure cultures of bacteria isolated from non-selective medium. Nearly half (43.5%; 214/492) were associated with the genus Pseudomonas. Five of the lakes (ALT, GRU, GOS, GOL, and WOR) were all distinguished by high levels of Pseudomanas (p ≤ 0.001). HIN, the highest elevation lake, had the highest percentage of ice+ strains. Our work highlights the potential for uncovering new haplotypes of P. syringae in aquatic habitats, and the use of robotic technologies to sample and characterize microbial life in remote settings.

Introduction

Bacteria in the genus Pseudomonas are ubiquitous in natural and managed environments (Berge et al., 2014). Members of the Pseudomonas syringae complex have been cultured from clouds, rain, snow, streams, rivers, and lakes. Some strains of P. syringae express an ice nucleation protein (hereafter referred to as ice+) that catalyzes the heterogeneous freezing of water (Morris et al., 2014). The ice+ phenotype has been used to compare P. syringae populations in a freshwater lake in Virginia, USA during different seasons (Pietsch, Vinatzer & Schmale, 2017). The P. syringae complex (sometimes referred to as P. syringae sensu lato (Bull et al., 2011)) is comprised of at least 15 recognized species and more than 60 different pathovars of P. syringae sensu stricto (Gomila et al., 2017; Gutiérrez-Barranquero, Cazorla & De Vicente, 2019). In this work, we use the name P. syringae to refer to members of the complex.

Benson et al. (2019) reported a new method to collect microorganisms from freshwater lakes in Austria using a DrOne Water Sampling SystEm (DOWSE). Briefly, a sterile conical 50 mL tube was inserted into a custom 3D-printed water sampler and carried by the drone on a 4.5 m long nylon tether to target locations in each lake. Samples were collected by drone from three different distances from shore (1, 25 and 50 m), and microorganisms were cultured on two types of agar media (one was semi-selective for Pseudomonas spp., and the other was a general growth medium). Here, we expand the work of Benson et al. (2019) by analyzing the haplotype diversity (based on the partial sequence of the citrate synthase housekeeping gene used to determine the phylogenetic context of strains; Berge et al., 2014) and ice nucleation activity of strains collected during the drone-based water sampling missions in eight lakes in Austria. It is important to note that the Benson et al. (2019) study focused on drone-based water sampling as a new method and approach for aquatic microbiology.

Based on the preliminary observations in Benson et al. (2019), we hypothesized that (1) the haplotype structure of P. syringae and other culturable bacteria varies among the eight Austrian lakes and (2) greater numbers of ice-nucleating strains of P. syringae are observed in colder Austrian lakes at higher altitudes. To test these hypotheses, strains from Benson et al. (2019) were sub-cultured, sequenced, subjected to haplotype analyses, and tested for ice nucleation activity at −8 °C. The specific objectives of this study were to: (1) investigate the haplotype structure of strains of P. syringae from the eight Austrian lakes based on partial sequences of the citrate synthases (cts) gene, (2) determine the ice nucleation activity of strains of P. syringae from the eight Austrian lakes using a droplet freezing assay, (3) examine the diversity of total culturable bacteria from the eight Austrian lakes based on partial sequences of 16s rDNA, and (4) investigate potential associations of populations of culturable bacteria among the eight Austrian lakes using a non-metric multidimensional diversity analysis (NMDS). Since the bacteria analyzed in this study were collected with drones, our work highlights the potential for robotic systems to monitor the diversity and life history of microorganisms in lake environments, particularly in remote alpine settings.

Materials and Methods

Lake locations and sampling

Freshwater sampling was conducted at eight different lakes in Austria in June 2018, Altauseer See (ALT), Grundlsee (GRU), Toplitzsee (TOP), Vorderer Gosausee (GOS), Gosaulacke (GOL), Hinterer Gosausee (HIN), Ossiacher See (OSS) and Wörthersee (WOR), (Fig. 1) (Benson et al., 2019). Sampling details and design of the Drone Water SamplEr (DOWSE) were previously described in Benson et al. (2019). Briefly, seven of the eight lakes were sampled with a drone 1, 25, and 50 m from the shoreline. Samples were collected from a 30 × 50 m grid of nine collection sites in the lake. Due to the narrow width of GOL, this lake was sampled at 1 and 25 m from shore in four locations to form a 45 × 25 m grid of eight sample collections (Benson et al., 2019). Lake area (km2) and elevation are listed in Table 1.

Figure 1 Map showing eight lakes sampled in Austria.

ALT, GRU, and TOP were sampled on field day 1, GOS, GOL, HIN on field day 2, and OSS and WOR on field day 3. These lake groups are shown as blue, red, and green label location pins, respectively. Map data © 2020 Google.

Table 1 Lake name, abbreviation, GPS location, size, altitude, depth at 50 m from shore, and temperature for all lakes sampled during the campaign.

Lake name	Abbreviation	Latitude and longitude	Approximate
area
(km2)	Elevation
above sea level
(meters)	Depth 50 m from shore (m)	Temperature
(C)	
Altauseer See	ALT	47°38′27.3″N 13°47′04.2″E	2.1	712	3.4	21	
Grundlsee	GRU	47°37′51.1″N 13°51′23.2″E	4.1 (5)	708	21.3	19	
Toplitzsee	TOP	47°38′30.0″N 13°55′40.0″E	0.54	718	16	22	
Vorderer Gosausee	GOS	47°31′28.1″N 13°30′52.7″E	0.67	917	16.2	21	
Gosaulacke	GOL	47°30′58.3″N 13°31′42.8″E	0.04	978	4.2	16	
Hinterer Gosausee	HIN	47°30′07.8″N 13°33′00.1″E	0.31	1,151	11.2	17	
Ossiacher See	OSS	46°39′54.7″N 13°58′01.8″E	10.8	502	8.6	25	
Wörthersee	WOR	46°37′19.6″N 14°09′06.0″E	19.4	440	24.8	26	

Lake sampling permissions and safety of drone operations

Permissions to sample the lakes were granted by the Austrian Federal Forests AG, DI Martin Heinz Stürmer, on 3 April 2018. A formal field collection permit was not required for this work. Sites for drone operations were carefully selected to be minimally intrusive to people in the area during the time of sampling. The drones used as part of this work were registered with the Federal Aviation Administration (FAA). The UAS pilot for the missions reported in this manuscript was a certified FAA Remote Pilot under Part 107, Certificate Number 4038906.

Collection and processing of microorganisms

Lake samples from Benson et al. (2019) were processed for storage on the same day of collection. Briefly, 100 mL from each location was filtered through a 0.2 µm single use Pall filter funnel (#28143-542; VWR International, Radnor, PA, USA). Filters were transferred to 15 mL tubes and stored at 4 °C. All samples were shipped on ice to VA, USA for culturing, storage, and downstream analyses. Details concerning the culturing of microorganisms for this study are described in Benson et al. (2019). Briefly, KBC agar plates were used to select for bacteria in the genus Pseudomonas and TSA agar plates were used to grow culturable bacteria (including Pseudomonas). While not all bacteria collected from the environment are culturable in a laboratory setting, we chose an agar plate composition that had been tested with precipitation samples. Previous studies explored various agar plate compositions (including TSA) with regard to the culturability of environmental rain (Failor et al., 2017), and simulated rain samples (Hanlon et al., 2017).

Sequence analysis and identification ( cts and 16S)

Pure strains were subjected to sequence analyses of the citrate synthase (cts) housekeeping gene (for strains on the semi-selective medium, KBC) or 16S rDNA (for strains cultured on TSA). Berge et al. (2014) used multi locus sequence typing (MLST) analysis of 216 P. syringae strains to identify 23 clades and describe 13 phylogroups in the P. syringae genetic complex. These authors showed that the cts housekeeping gene alone was 97% accurate in predicting phylogeny. Prior to sequencing, a 5 min. template boil was performed in 100 µL of water with a 1 mm toothpick stab of frozen stock or from a fresh colony streak plate. A 25 µL PCR reaction was carried out with 2 µL of template from the quick boil tube. The cts gene was amplified using GoTaq® Green Master Mix (Promega M712) with primers ctsFor (5′ CCC GTC GAG CTG CCA ATW CTG A 3′) and ctsRev (5′ ATC TCG CAC GGS GTR TTG AAC ATC 3′). The cts gene sequence data was used to confirm that strains belong to the P. syringae complex and to determine the phylogroups within the complex. Conditions of the cts PCR reaction were previously described in Pietsch, Vinatzer & Schmale (2017). PCR products were confirmed on an 1% TBE agarose gel with SYBRTM Safe DNA Gel Stain (Invitrogen by Thermo Fisher Scientific, Waltham, MA, USA). Reactions were cleaned enzymatically with rSAP and ExoI and Sanger sequencing reactions were performed by Eton Biosciences (AB1 3730 × 1 DNA Sequencer; Eton Biosciences, Durham, NC, USA). Sanger generated sequences were trimmed (using Phred Version 0.020425.c; http://bozeman.mbt.washington.edu/phrap.docs/phred.html) and subsequently queried against the NCBI nr (non-redundant) database. The same PCR parameters were used to amplify a portion of the 16S rDNA gene. The forward and reverse primers used were 518 F (5′ CCA GCA GCC GCG GTA ATA CG 3′) and 1491 R (5′ ATC GGY TAC CTT GTT ACG ACT TC 3′) (Turner et al., 1999). The identification of culturable bacteria was determined to the classification level of genus by using trimmed sequences as described above and sequences were submitted to GenBank.

Phylogenetic analysis of P. syringae complex strains in lakes

To specifically determine which of the strains isolated from KBC medium were within the P. syringae complex and to situate them phylogenetically, we compared, via BLAST, the cts sequences for these strains with the overall NCBI data base which contains the full diversity of this bacterium reported by Berge et al. (2014). Strains for which the cts sequence did not have at least 90% sequence similarity with that of strains in the P. syringae complex were eliminated from subsequent analysis thereby facilitating alignment of the cts sequences along a greater number of bases than if all strains were included in the analysis. Four additional strains were eliminated because we could not sequence the cts gene to obtain a sufficient number of bases (>300). The resulting numbers of P. syringae complex sequences for each lake were 10 for ALT, 30 for GRU, 37 for TOP, 39 for GOS, 14 for GOL, 18 for HIN, 39 for OSS and 1 for WOR. The phylogenetic relationships among strains was determined as previously reported (Berge et al., 2014) based on 311 base pairs of the retained sequences and for 34 reference strains via the neighbor-joining algorithm in MEGA7 with 1,000 bootstrap replicates. The total number of confirmed P. syringae complex strains from each lake is shown in Table 2.

Table 2 Lake name abbreviation, collection date, number of confirmed strains belonging to the P. syringae complex from each lake, number of strains assayed for ice+ (163 of 188), number of frozen samples, and percent of frozen strains confirmed to belong to the P. syringae complex.

Abbreviation	Date of collection	Confirmed as part of the P. syringae complex based on HT analysis	Strains tested for ice+	Total ice+	Percent ice+	
ALT	June 7 2018	10	10	1	10%	
GRU	June 7 2018	30	30	13	43%	
TOP	June 7 2018	37	28	8	29%	
GOS	June 8 2018	39	30	13	43%	
GOL	June 8 2018	14	13	4	31%	
HIN	June 8 2018	18	17	13	76%	
OSS	June 10 2018	39	34	13	38%	
WOR	June 10 2018	1	1	0	0%	
All lakes	Total	188	163	65		

Ice-nucleation activity of confirmed P. syringae complex strains

Confirmation of positive ice+ Pseudomonas samples that were tentative P. syringae (271 strains) was verified in three independent ice-nucleation assays and reported in Table S1. Positive ice+ strains that were part of the P. syringae complex are listed in Table 2. The assays were performed on a boat made with PARAFILM® M (Sigma P6543, 20 in. × 50 ft) floating in a Lauda Alpha RA 12 (LCKD 4908) cooling bath (08075; LAUDA-Brinkmann, LP, Delran, NJ, USA) with ethylene glycol coolant fluid (Air gas RAD64000246). This parafilm boat-based assay has been described previously (Pietsch et al., 2016; Garcia et al., 2019). Pure cultures of bacteria were transferred from culture plates with a sterile toothpick to 140 µL of DI water into a 96-well plate. Plates were vortexed for 30 s and incubated at 4 °C for 1 h. Twelve-microliter droplets of the microbial suspension were loaded in duplicate at −6 °C. The temperature of the cooling bath was decreased to −7 °C, held for 2 min, then decreased to −8 °C. After a final incubation time of 10 min at −8 °C, all drops that froze were considered to be INA+ samples. The freezing temperature was recorded for each drop on the cryofloat.

Analysis of microbial richness and similarity

16S rDNA sequences were analyzed from a total of 492 pure cultures of bacteria isolated from TSA. Richness and Shannon diversity were characterized at all levels of taxa for 492 strains cultured on general enriched TSA agar. Variation in richness and diversity with distance to shorelines and lakes levels were analyzed with mixed effect models with lake treated as a random effect. Random effects were tested using likelihood ratio tests whereas fixed effects were tested using F-tests with Satterthwaite adjusted degrees of freedom.

Population analyses

Variation in the composition of genera of bacteria among and within lakes was analyzed using non-metric multidimensional scaling (NMDS) (Shepard, 1962a, 1962b; Kruskal, 1964a, 1964b) and PERMutational Multivariate ANOVA (PERMANOVA; Anderson, 2001). Dissimilarity matrices for NMDS and PERMANOVA were created using Bray-Curtis dissimilarity (Bray & Curtis, 1957). The resulting 2D NMDS ordination was adequate for drawing inferences (stress = 0.14). All analyses were performed using the R statistical environment (R core team 2019) with heavy reliance on the packages vegan (Oksanen et al., 2019), lme4 (Bates et al., 2015), lmerTest (Kuznetsova, Brockhoff & Christensen, 2017), and asbio (Aho, 2023).

Results

Culturing and identification of strains in the P. syringae complex

A total of 415 strains recovered from KBC were identified as Pseudomonas. Sequences were submitted to GenBank and assigned accession numbers (MW857572–MW857985 and MW892633). Based on specific sequence criteria relative to the cts gene, 271 strains were defined as tentative P. syringae based on significant sequence homology. These strains were then analyzed using a Pseudomonas specific database to determine if they were part of the P. syringae complex defined by Berge et al. (2014). A total of 188 strains were associated with the P. syringae complex and represented 32 haplotypes in eight phylogroups (Fig. 2). Workflow is shown in Fig. S1.

Figure 2 Haplotype diversity of 188 strains in the P. syringae complex from eight lakes in Austria.

Solid colored bars indicate the fraction of haplotypes (HT) in each lake. Colors represent the haplotypes of different phylogroups (PG) that were in common to at least two lakes. Haplotypes in black (a total of 21 HT) were only present in one of the eight lakes. Hashed bars indicate the total population concentration of colonies selected on KBC in each lake (103 cfu/mL).

The numbers of tentative P. syringae sequences for each lake were 26 for ALT, 53 for GRU, 53 for TOP, 41 for GOS, 24 for GOL, 21 for HIN, 49 for OSS and 4 for WOR for a total of 271 strains. A phylogenetic analysis of these strains showed that 188 strains grouped with P. syringae as part of the P. syringae complex, and the remaining 83 grouped with reference strain PAO1 for P. fluorescens. The 188 strains in the P. syringae complex represented 32 haplotypes in phylogroups PG01, PG02, PG07, PG09, PG10, PG13, PG14 and PG15 of the P. syringae complex, for the portion of the cts gene that was sequenced. Among the 32 haplotypes, 11 were common to at least two lakes and five lakes had a total of 21 unique haplotypes, i.e., haplotypes represented by only one strain in one lake (Fig. 2).

The structure and size of the P. syringae complex phylogroups varied among lakes, and we found few correlations among the factors measured here. For example, neither the concentration nor the abundance of any of the haplotypes or phylogroups were significantly correlated with lake altitude (Spearman’s rank correlation, p < 0.05). PG01 and PG02 were the only phylogroups whose fractions in the population were significantly correlated with total concentrations of P. syringae complex strains in lakes; they increased as total concentrations increased (Spearman’s rank correlation, p < 0.05).

Ice nucleation activity of P. syringae strains

The numbers of tentative Pseudomonas strains that were tested for ice nucleation activity at −8 °C (and % that froze) in the droplet freezing assay for each lake are included in Table S1. There were 26 for ALT (12%), 50 for GRU (38%), 44 for TOP (18%), 40 for GOS (35%), 23 for GOL (26%), 20 for HIN (65%), 44 for OSS (30%) and 4 for WOR (0%) (Table S1). The frequency of tentative Pseudomonas strains that were ice nucleation active (at −8 °C) varied among lakes (0 to 65%) and was significantly positively correlated with the fraction of phylogroup 02 (PG02) strains in the population but with no other phylogroup component nor with the total abundance of tentative Pseudomonas strains in the lake (Spearman’s Rank correlation, p < 0.01).

Microbial richness and similarity among the eight lakes

Thirty-five unique genera of culturable bacteria were identified from the non-selective TSA plates across the eight lakes (7.34 × 104 cfu/mL mean of eight lakes). These strains were assigned accession numbers MN490845 to MN491336. Over forty percent of the strains (43.5% (214/492)) were associated with the genus Pseudomonas. Population richness (based on genera of bacteria) and Shannon diversity were analyzed for the set of 492 bacteria identified from the eight lakes. The highest richness was detected in TOP and OSS lakes, while the lowest richness was reported in GOS and GOL (Fig. 3). Pseudomonas was the most abundant genus, and was present in all eight lakes at a mean percentage of 43.5% (214/492). The percentage of Pseudomonas ranged from 23% in HIN, to 64% in GOS. ALT, GRU, GOS, GOL and OSS had 50% or more Pseudomonas in the strains analyzed (Fig. 4). The Shannon diversity analysis confirmed TOP and OSS to have the highest diversity. The analysis also showed more diversity in HIN relative to ALT, whereas the richness data did not distinguish a difference between HIN and ALT.

Figure 3 Richness (A) and Shannon diversity (B) of lakes.

Lake locations were significantly different with respect to both richness (4.41, p = 0.036) and Shannon diversity (4.2, p = 0.04) after controlling for distance to shoreline. The trend for level of richness and diversity was the same for the eight lakes. Error bars are SEMs.

Figure 4 Bar graph of diversity at the classification level of genus for microbial growth on general growth TSA media (total 492 sequences with 35 unique genera identified).

P. syringae was the only microbe present in all eight lakes. Taxa in bar plots are stacked from most abundant (at the bottom of bars) to least abundant (at the tops of bars), and taxa names in legend also follow this convention.

Population analysis of culturable bacteria

A PERMANOVA of samples within lakes indicated that bacterial assemblages varied strongly by lake (R2 = 0.58, F2,5 = 2.81, p = 0.001, 999 permutations). Bacterial assemblages did not vary with distance to lake shoreline (R2 = 0.04, F2,5 = 0.73, p = 0.745, 999 permutations).

An NMDS analysis showed relationships of bacterial assemblages within and among lakes (Fig. 4). Particularly distinct were assemblages of TOP (lower right), HIN (center right) and OSS (top). In the 2D ordination for genus, the vector fitting p-values less than or equal to 0.004 (p ≤ 0.004) showed the strongest association to the ordination as marked by arrows in Fig. 5. ALT, GRU, GOS, GOL, and WOR were all distinguished by high levels of Pseudomonas (p ≤ 0.001). WOR also showed a correlation to Exiguobacterium (p ≤ 0.036). TOP was indicated by Janthinobacterium (p ≤ 0.001), while OSS was indicated by Pedobacter (p ≤ 0.001) and Brevundimonas (p ≤ 0.003). HIN was correlated to two genera, Flavobacterium (p ≤ 0.015) and Massilia (p ≤ 0.016).

Figure 5 Two dimensional NMDS projection of lake genera data (stress = 0.14).

Vector fitting of genera with significant correlations with the projection p-values (p ≤ 0.04) are marked by arrows. The arrows point toward the region of most rapid increase in the projection and arrow length is scaled by the strength of the correlation.

Discussion

Knowledge of microbial ecology and functions in aquatic systems, and in particular alpine lakes, can be hampered by the difficulty to access lakes for sampling. Benson et al. (2019) reported a new method and approach to collect microorganisms from freshwater lakes in Austria using a Drone Water Sampling SystEm (DOWSE). Here, we expand the work of Benson et al. (2019) through a comprehensive analysis of the diversity and ice nucleation activity of strains collected during those drone-based missions. The work presented in this manuscript expands our understanding of the aquatic microbiology of remote alpine lakes in Austria, and adds to a growing body of literature on the biogeography and ubiquity of Pseudomonas in aquatic habitats (Christner et al., 2008; Morris et al., 2008, 2010; Pietsch, Vinatzer & Schmale, 2017; Morris et al., 2022a).

High levels of Pseudomonas in five of the lakes supports the ubiquitous nature of P. syringae in aquatic environments. The set of 188 strains confirmed to belong to the P. syringae complex represented 32 haplotypes in eight phylogroups. Phylogroup 02 (PG02) is the most ubiquitous group, and includes strains that have been collected from a range of habitats, including rain, trees, and an irrigation basin (Berge et al., 2014). For our dataset, 43% (80/188) of the identified strains belong to PG02 and were collected from six of the eight lakes. The ice+ phenotype was reported at 85% for non-pathogenic PG02 environmental strains tested by Berge et al. (2014). In our study, PG09 and PG07 were represented by 45 and 27 strains, respectively. PG09 is exclusively composed of aquatic habitat collections while PG07 represents strains responsible for pathogenicity in potato (Berge et al., 2014). PG13 was the fourth most abundant pathogroup with 24 strains represented by three lakes (OSS, GRU, TOP). Interestingly, PG13 is defined by strains isolated from non-plant sources, from wild alpine plants (Berge et al., 2014) and from five Icelandic habitats (Morris et al., 2022b).

Seven of the eight lakes had tentative P. syringae strains that were ice+. Lake locations can be seen in Fig. 1, and lake characteristics including size, depth, and elevation can be seen in Table 1. Our hypothesis that samples from higher elevations would contain more ice+ bacteria was supported by our data; the lake with the highest elevation (HIN) had the most ice+ strains of all the lakes. HIN was also one of the coldest lakes at 17 °C, only one degree warmer than GOL, the coldest lake. It is worth noting that GOL is only 0.04 km2 in area, so temperature changes might be observed at different rates in smaller lakes such as GOL compared to larger lakes such an HIN (area of 0.31 km2). Two of the three deepest lakes (GRU and GOS) had over 40% ice+ strains. These lakes were also more than double the altitude of WOR, the deepest lake and at the lowest elevation of the lakes studied.

A PERMANOVA of samples within lakes indicated that bacterial assemblages varied strongly by lake. This variation among lakes supports the notion that microbe composition is linked to sample source and immediate environmental conditions, including land-use of the surrounding area (Bowers et al., 2011). GOS, GOL, and HIN are located close to each other in Upper Austria (Fig. 1). The straight-line distance between GOS and HIN is 4 km with GOS and GOL below 1,000 m in altitude and HIN above 1,100 m (Table 1). HIN had a greater bacterial diversity than GOS and GOL.

Microbial assemblages did not vary with distance from shoreline (R2 = 0.04, F2,5 = 0.73, p = 0.745, 999 permutations). However, high concentrations of microbes were observed in surface “hot spots” within a 30 m × 50 m sampling grid on the eight lakes (Benson et al., 2019). A “hot spot” on the surface included one or more of nine sampling locations in the 150 m2 sampling grid that had high concentrations of microbes. Wind across these hotspots could engender aerosolization and transport of microbes at elevated frequencies (Pietsch et al., 2018). Moreover, improved imaging and tracking techniques could be utilized in future work to predict hot spots or guide collection efforts to the center of the hot spots, such as visible scum on the water surface associated with high concentrations of toxic cyanobacteria commonly referred to as harmful algal blooms (HABs) (Tian et al., 2017; Ma et al., 2021).

Supplemental Information

Supplemental Information 1 Data for haplotype assignments and accession numbers.

Click here for additional data file.

Supplemental Information 2 Selection and identification scheme of Pseudomonas.

(1) Bacterial growth shown for a TSA and KBC plate from Lake 1 (ALT). Colonies from KBC became the set of tentative Pseudomonas stains. (2) GenBank accession of MW857586 from the set of 415 confirmed Pseudomonas strains that were verified by partial sequencing of the cts gene. (3) Example of a sequence alignment of tentative P. syringae strains. The highlighted strains, MW857586.1 and MW857595.1, grouped with P. syringae, while MW857579.1 did not. (4) Visual of phylogeny example of P. syringae strains mapping to different phylogroups in the P. syringae complex.

Click here for additional data file.

Supplemental Information 3 Ice nucleation assays for tentative strains of P. syringae.

Lake name abbreviation, collection date, number of tentative P. syringae strains from each lake, number of tentative P. syringae strains assayed for ice+ (251 of 271), total number of tentative P. syringae ice+ frozen samples, percent of frozen ice+ strains from tentative P. syringae.

Click here for additional data file.

Additional Information and Declarations

Competing Interests

Author Contributions

Field Study Permissions

Data Availability

The authors declare that they have no competing interests.

Regina Hanlon conceived and designed the experiments, performed the experiments, analyzed the data, prepared figures and/or tables, authored or reviewed drafts of the article, and approved the final draft.

Celia Jimenez-Sanchez conceived and designed the experiments, performed the experiments, analyzed the data, prepared figures and/or tables, authored or reviewed drafts of the article, and approved the final draft.

James Benson conceived and designed the experiments, performed the experiments, authored or reviewed drafts of the article, and approved the final draft.

Ken Aho conceived and designed the experiments, performed the experiments, analyzed the data, prepared figures and/or tables, authored or reviewed drafts of the article, and approved the final draft.

Cindy Morris conceived and designed the experiments, performed the experiments, analyzed the data, prepared figures and/or tables, authored or reviewed drafts of the article, and approved the final draft.

Teresa M Seifried conceived and designed the experiments, performed the experiments, authored or reviewed drafts of the article, and approved the final draft.

Philipp Baloh conceived and designed the experiments, performed the experiments, authored or reviewed drafts of the article, and approved the final draft.

Hinrich Grothe conceived and designed the experiments, performed the experiments, analyzed the data, authored or reviewed drafts of the article, and approved the final draft.

David Schmale conceived and designed the experiments, performed the experiments, analyzed the data, prepared figures and/or tables, authored or reviewed drafts of the article, and approved the final draft.

The following information was supplied relating to field study approvals (i.e., approving body and any reference numbers):

Martin Heinz Stürmer (Österreichische Bundesforste AG) and the Austrian Federal Forests AG provided the support and permissions to sample the eight lakes described in this article.

Lake sampling permissions and safety of drone operations

Permissions to sample the lakes were granted by the Austrian Federal Forests AG, DI Martin Heinz Stürmer, on 3 April 2018. A formal field collection permit was not required for this work. Sites for drone operations were carefully selected to be minimally intrusive to people in the area during the time of sampling. The drones used as part of this work were registered with the Federal Aviation Administration (FAA). The UAS pilot for the missions reported in this manuscript was a certified FAA Remote Pilot under Part 107, Certificate Number 4038906.

The following information was supplied regarding data availability:

The 1,563 GenBank accession numbers are available in Table S1.

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
