# Peer review of "Diversity and ice nucleation activity of Pseudomonas syringae in drone-based water samples from eight lakes in Austria"

_PeerJ, doi:10.7717/peerj.16390_

## Round 0.1 · original submission · Major Revisions

All of the reviewers found your manuscript to be interesting but raised multiple issues that need to be addressed. Please pay careful attention to the comments of Reviewers 1 and 2, including the annotated manuscript supplied by Reviewer 2. All reviewers also raised issues of clarity in their reviews, so please consider revising and clarifying the writing throughout. Thank you for your submission to PeerJ.

Reviewer 1 ·

Basic reporting

This manuscript has added incrementally to our understanding of the distribution of ice nucleation active bacteria in the world. Specifically, the study has followed up on a study reported in 2019 by Benson at al that has used drones to sample lake water. Throughout the manuscript, there is frequent reference to the Benson study, but it was never clear to me whether the samples analyzed in this study were actually collected in the Benson study. That is, I kind of got the impression that Benson described an airborne device that could collect samples, but it wasn't clear what was concluded from any samples that were collected by Benson. In a couple of locations in this new manuscript , they seem to refer to the fact that ice nucleating bacteria were also described in the Benson study. This new submission absolutely needs to provide more background on what was reported in the Benson study and how this study differs from it. For the most part, this new submission seems to have been done carefully and provides a wealth of information about the microbiological content of the Austrian lakes that were sampled. I was surprised and impressed that Pseudomonas species were so dominant in these lakes, and also that a relatively high percentage of the Pseudomonads are ice nucleation active. I was also a bit surprised that the total cultural bacteria were so low. I had always the impression that lake waters had populations in excessive 10,000 cells per ml or more, but that seem to have been high relative to what was seen here. The authors might want to comment on this. Were these lakes really that pristine? The biggest issue that I see with the manuscript, is that a number of the statements are very awkwardly written and often either make no sense or are ambiguous. The organization of the manuscript was also not as tight as I might have expected. For example, the discussion was quite long, and included a lot of material that I would have thought should have been more concisely summarized in the results section. Several points seem to have been revisited several different times both in the results and the methods, making me think to myself “haven't I just read this elsewhere?”. I will attempt to note some of these in the more specific comments below. After substantial editing to tighten up the language and probably also to reorganize the way the data is described and subsequently discussed, the manuscript will provide new information further showing that ice nucleating Pseudomonads are quite ubiquitous in the environment, especially aquatic environments.

Specifics:
Line 24. I have no idea what they mean when they refer to “subjected to a series of population analyzes”. This needs to be made less obtuse.

Lines 32 and 33. The reference to “portions of 16S sequences” is awkward. It should read something like “partial 16s sequences”, or more precisely to refer to the variable region that was sequenced.

Lines 38 through 39. The reference to “the level of genus” is very awkward. I have no idea what they are talking about.

Lines 39 through 44. The description of the dominant taxa is done in a way that makes it very hard to understand. This is very awkward.

Line 54. I really didn't like their abbreviation “P.spp” to refer to Pseudomonas species. I have never seen such an abbreviation before, and the few characters they saved by doing this here is not worth the aggravation to readers. Simply say “Pseudomonas spp.”.

Lines 78-79. Not only do I not know what they mean when they say their study “used a bona fide P. syringae data set”, but I don't see how that could actually have helped them “determine the ice plus phenotype”. This needs to be made clear.

Line 86. I don't understand when they seem to say that these random hotspots were “ significantly different for all lakes”. This doesn't make sense.

Line 95. It would have been very helpful to this reviewer to have given a bit more background about how the sequence analysis of the citrate synthase gene was it self-sufficient to be defining the phylogroups of Pseudomonas syringae. I am not an expert in the genotyping of Pseudomonas, and this may well be the appropriate way (I would have thought that MLST would have been used). Given that professor Morris was one of the authors of the study more details should be available - and more literature citations to justify this method would have been helpful.

Line 129. Here and elsewhere, it is not at all clear whether the bacteria that was analyzed here were actually cultured and sampled from the Benson study.

Line 140 through 141. This sentence is not clear. I have no idea what they are trying to say.

Lines 141 through 145. I really don't understand the relevance of this statement of some other results of other study that has been placed here in the methods section. This seems like it could be eliminated.

Lines 189 through 192. Again, the way this is written, it would seem that this samples described in this current submission or actually collected and described previously in the Benson study. This needs to be made clear whether it is true or not.

Lines 196 through 199. I am a bit surprised that they made boats out of parafilm itself rather than using paraffin coated aluminum that has been more typically used in such droplet freezing assays. I would have expected that the paraffin would act as an insulator and that the droplets may not be as close to the temperature of the ethylene glycol coolant as they would have been on an aluminum surface. This may not be fatal, but did surprise me.

Lines 201 through 205. The way this method is described, it would seem to suggest that they froze only two droplets. I would hope this is not correct. They definitely need to make this clearer. If in fact they only drop froze 2 droplets I would be very disappointed, given that there is a decent possibility of spontaneous freezing of some of the droplets, and this might have given them false positive responses. While the frequency of nucleation in control droplets that would not contain ice+ bacteria is quite low, with such small numbers of droplets, they might well have misled themselves in some cases if only two drops were tested.

Lines 225 to 227. It is unclear what they mean by “putative P. syringae”. Did they previously think they were P. syringae and then ended up grouping with Pseudomonas fluorescence.? To refer to the “83 branched with the reference strain” is very awkward. Certainly “branch” is not the right word – “grouped is better.

Line 256. I really don't know what they mean by “compositional resemblance”.

Lines 290 to 292. The sentence is very awkwardly written. Instead of saying that phylogroup 02 “comprised of collections” from various habitats, it would be much more accurate to say that phylogroup 02 “include strains that have been collected from these various habitats”. The same issue is also present in their description of the other phylogroups.

Lines 301 through 315. I found this long discussion comparing their results to that of a previous study done in Virginia to be very awkward and unnecessary. I didn't see that it added much - and it could have been summarized in a single sentence saying that a similar frequency of ice+ P. syringae was observed as in virginia.

Line 322 through 324. I don't understand the logic in this sentence. Furthermore, towards the end of the discussion they have extensively referred to the likelihood that the surrounding vegetation and land use etc was contributing to the differences in the numbers and types of bacteria seen in these lakes. This seems very likely, but at the same time, readers were not given any information that would help us get a better feel for whether this is likely. That is, a table or some other summary of the features of the lakes and the surrounding area would definitely have helped readers make more conclusions about how to interpret the results of this study. I would strongly urge additional information to be provided about the surrounding area and hence the meta community that probably led to the microbes seen in the water.

Line 330 through 331. This very awkward sentence is very hard to understand.

Lines 332-335. Again, a more detailed description of the lakes in the surrounding region would be very helpful here.

Lines 336 through 345. This material all belongs in the results, and not in the discussion, since there is no discussion of it here.

Line 347-348. This sentence is very awkward. I don't understand it.

Line 358 through 359. I am still having trouble understanding what they refer to as a “ hotspot”. Has such localized concentration of bacteria in open waters been described by others? In a freely mixed body of water like a lake, it is hard to imagine how it could occur. Also, I don't quite understand how the whole vectoring story applies only to a hotspot. I would assume that the processes such as wind splash etc that would release bacteria from a lake would occur equally likely at any location on the surface of the lake, and not only at the hotspot.

Lines 363 through 367. I don't understand any of this. I don't quite see how looking at a surface of water is going to give us much information about what was going on within the water body itself.

Lines 369 through 384. The conclusions section seems to simply repeat everything we've already read earlier. It can easily be dropped.

Experimental design

See basic reporting

Validity of the findings

See basic reporting

Additional comments

See basic reporting

Reviewer 2 ·

Basic reporting

The manuscript presents data associated with sampling via drones of lakes at different elevations in Austria to identify what is referred to as Bona fide Pseudomonas syringae populations. Although the introduction describes previous literature relating to sampling from noncrop locations previously, there is a need for basic information on Pseudomonas syringae leading up to the work. For instance phylogroup is used but what is the relevance of phylogroup in this manuscript. Secondly, the authors use bona fide to describe P. syringae. The authors should also introduce sensu lato and stricto sensu given they are using P. syringae in a very broad sense.

Experimental design

The experimental design seems to be fine except that it would help to provide information on the method of sampling via drones.

Validity of the findings

It is difficult to assess some of the findings as they are not presented as best I can see. Thirty-two haplotypes were identified in the manuscript although these are not obvious in the manuscript. Why not include a supplementary file so reader can judge. In the results in figure 2 four phylogroups are shown although in results eight pgs are indicated. Why not include these in a supplementary table? Why is there not a Shannon's diversity index to make the points as to relative diversity. Other lakes also had a high number of genera. Rather than focusing on the diversity associated with elevation, it would be more interesting to focus more on the unique bacterial populations in three lakes indicated in the NMDS analysis.

Annotated reviews are not available for download in order to protect the identity of reviewers who chose to remain anonymous.

Reviewer 3 ·

Basic reporting

The manuscript from Hanlon et al. describes the isolation and characterization of water samples from a variety of Austrian lakes. The authors sample water, and plate out on either TSA or KBC media to isolate/identify total cultureable bacteria and potential Pseudomonas syringae strains (respectively). The authors also report on metrics and statistics concerning cultureable bacterial communities within the lakes and report on ice nucleation potential of Pseudomonads. There is an interesting relationship reported between elevation of the lake and ice nucleation percentage of the strains.

Overall, I felt that the data collection and analyses were adequate for publication. However, I also felt that the manuscript could be significantly streamlined and reconfigured to be more clear about the questions being tested and how the experiments addressed these questions.

Abstract: The abstract is far too long. Please condense and streamline this so that it reads as one/two paragraphs and does not include fine scale data and interpretations of the data. Please reshape the abstract so that it reflects quick take home messages of the manuscript.

Introduction/Discussion: I think these sections could be reconfigured and reshaped as well, to better reflect the questions being asked and how the experiments address the questions. The authors move back and forth between analyses of cultureable bacteria from the lakes to Pseudomonad percentages to ice nucleation percentages, but the intro and discussion as is reads as a bit too scattered and could be made much more linear and direct. The authors should also highlight why they are investigating community diversity in the context of Pseudomonads and what these analyses actually tell the reader. I went in thinking that this paper would be concise and report back on number of ice nucleating strains per lake (which it ultimately does, just in a roundabout way). Please include additional context and transition sentences and information that helps the reader to understand why the additional community level analyses and statistics are being reported.

Experimental design

I think the methods and analyses are quite clear, although the authors should include the number of times that the ice nucleation tests were performed (they should have at least been repeated). Please include information on replication of the ice nucleation assays prior to resubmission.

Validity of the findings

I think the results are pretty straightforward and reported as such. No other comments.

Additional comments

As I mentioned above, I think the authors should consider how to reconfigure the intro and discussion so that the take away message of the manuscript is clear and concise. I have no problem with descriptive papers, I just want to be able to capture the take home message after reading.

---

## Round 0.2 · accepted · Accept

Two of three previous reviewers reviewed the revised manuscript and have confirmed that all previous comments were addressed. They have no further comments and agree that the manuscript is now ready for publication in PeerJ - congratulations!

Reviewer 1 ·

Basic reporting

The authors have been rather responsive to the comments made by myself and the other two referees. The increase in clarity of the massachusetts, and the history and organization of the study is now adequate. I am quite satisfied with the manuscript as revised.

Experimental design

See above

Validity of the findings

See above

Additional comments

None. Much improved and fine as revised.

Reviewer 3 ·

Basic reporting

The manuscript now meets all standards set out in the review template.

Experimental design

The manuscript now meets all standards set out in the review template.

Validity of the findings

The manuscript now meets all standards set out in the review template.

---

## Author Rebuttal · Round 0.2

**David G. Schmale III, Ph.D.**
Professor
403 Latham Hall, Ag Quad Lane
Virginia Tech
Blacksburg, VA 24061-0390
Phone: 540/231-6943
Fax: 540/231-7477
E-mail: dschmale@vt.edu

September 7, 2023

Editor, *PeerJ*

Dear Editor,

Attached please find our revised manuscript titled '**Diversity and ice nucleation activity of *Pseudomonas syringae* in drone-based water samples from eight lakes in Austria**' to be considered for publication in *PeerJ*.

The revised manuscript and associated materials have been uploaded through the online manuscript submission and peer review system. Changes have been 'tracked' in MS Word. A separate file containing our detailed responses to the three reviewers has also been included following this letter.

The abstract has been reduced significantly. The introduction has been re-organized and re-focused to clarify that the strains studied in this paper were from the Benson et al. (2019) drone-sampling missions in Austria. Additional methods have been added for the drone-based water sampling, and the section on the ice nucleation assay has been simplified. The language has been updated throughout regarding our population analyses and the scientific nomenclature referring to the *P. syringae* complex; the *P. syringae* complex (sometimes referred to as *P. syringae sensu lato* (Bull et al., 2011)) is comprised of at least 15 recognized species and more than 60 different pathovars of *P. syringae sensu stricto* (Gomila et al., 2017; Gutiérrez-Barranquero, Cazorla & De Vicente, 2019). We have added supplementary data for the haplotypes, and a supplementary figure to highlight our strain selections and analyses. The discussion has been significantly reduced to focus on the main points, and additional references have been added where appropriate.

Thank you for coordinating a thorough and detailed review of our paper. We look forward to having our paper published in *PeerJ*.

Sincerely,

David G. Schmale III, Ph.D.

*Please see our responses below in* **bold.**

==Reviewer 1==

Basic reporting

This manuscript has added incrementally to our understanding of the distribution of ice nucleation active bacteria in the world. Specifically, the study has followed up on a study reported in 2019 by Benson at al that has used drones to sample lake water. Throughout the manuscript, there is frequent reference to the Benson study, but it was never clear to me whether the samples analyzed in this study were actually collected in the Benson study. That is, I kind of got the impression that Benson described an airborne device that could collect samples, but it wasn't clear what was concluded from any samples that were collected by Benson. In a couple of locations in this new manuscript , they seem to refer to the fact that ice nucleating bacteria were also described in the Benson study. This new submission absolutely needs to provide more background on what was reported in the Benson study and how this study differs from it. For the most part, this new submission seems to have been done carefully and provides a wealth of information about the microbiological content of the Austrian lakes that were sampled. I was surprised and impressed that Pseudomonas species were so dominant in these lakes, and also that a relatively high percentage of the Pseudomonads are ice nucleation active. I was also a bit surprised that the total cultural bacteria were so low. I had always the impression that lake waters had populations in excessive 10,000 cells per ml or more, but that seem to have been high relative to what was seen here. The authors might want to comment on this. Were these lakes really that pristine? The biggest issue that I see with the manuscript, is that a number of the statements are very awkwardly written and often either make no sense or are ambiguous. The organization of the manuscript was also not as tight as I might have expected. For example, the discussion was quite long, and included a lot of material that I would have thought should have been more concisely summarized in the results section. Several points seem to have been revisited several different times both in the results and the methods, making me think to myself "haven't I just read this elsewhere?". I will attempt to note some of these in the more specific comments below. After substantial editing to tighten up the language and probably also to reorganize the way the data is described and subsequently discussed, the manuscript will provide new information further showing that ice nucleating Pseudomonads are quite ubiquitous in the environment, especially aquatic environments.

**Response: Thank you for your thorough review of the manuscript. We have overhauled the manuscript to address your suggestions and criticisms. The abstract has been reduced significantly. The introduction has been re-organized and re-focused to clarify that the strains studied in this paper were from the Benson et al. (2019) drone-sampling missions in Austria. The discussion has been significantly reduced to focus on the main points, and additional references have been added where appropriate.**

Specifics:
Line 24. I have no idea what they mean when they refer to "subjected to a series of population analyzes". This needs to be made less obtuse.
**Response: We have removed this language from the abstract.**

Lines 32 and 33. The reference to "portions of 16S sequences" is awkward. It should read something like "partial 16s sequences", or more precisely to refer to the variable region that was sequenced.
**Response: This has been changed to 'partial' as suggested.**

Lines 38 through 39. The reference to "the level of genus" is very awkward. I have no idea what they are talking about.
**Response: We have removed this language from the abstract.**

Lines 39 through 44. The description of the dominant taxa is done in a way that makes it very hard to understand. This is very awkward.
**Response: We have removed this section to improve clarity.**

Line 54. I really didn't like their abbreviation "P.spp" to refer to Pseudomonas species. I have never seen such an

abbreviation before, and the few characters they saved by doing this here is not worth the aggravation to readers. Simply say "Pseudomonas spp.".

**Response: We have changed this to "Pseudomonas spp." throughout the manuscript as suggested.**

Lines 78-79. Not only do I not know what they mean when they say their study "used a bona fide P. syringae data set", but I don't see how that could actually have helped them "determine the ice plus phenotype". This needs to be made clear.

**Response: We have removed 'bona fide' and changed the language of the manuscript to further define strains used in this study. We have also added additional language and references in the introduction to improve clarity on the *P. syringae* complex, and ice nucleation designations.**

Line 86. I don't understand when they seem to say that these random hotspots were " significantly different for all lakes". This doesn't make sense.

**Response: This language has been changed to avoid confusion.**

Line 95. It would have been very helpful to this reviewer to have given a bit more background about how the sequence analysis of the citrate synthase gene was it self-sufficient to be defining the phylogroups of Pseudomonas syringae. I am not an expert in the genotyping of Pseudomonas, and this may well be the appropriate way (I would have thought that MLST would have been used). Given that professor Morris was one of the authors of the study more details should be available - and more literature citations to justify this method would have been helpful.

**Response: We have added additional language in the methods section to address this:**
**Berge et al. (2014) used Multi Locus Sequence Typing (MLST) analysis of 216 *Pseudomonas syringae* strains to identify 23 clades and describe 13 phylogroups in the *Pseudomonas syringae* genetic complex. These authors showed that the citrate synthase (cts) housekeeping gene alone was 97% accurate in predicting phylogeny.**

Line 129. Here and elsewhere, it is not at all clear whether the bacteria that was analyzed here were actually cultured and sampled from the Benson study.

**Response: We have added additional language to address this:**
**Here, we expand the work of Benson et al. (2019) by analyzing the diversity and ice nucleation activity of strains collected during the drone-based water sampling missions in eight lakes in Austria.**

Line 140 through 141. This sentence is not clear. I have no idea what they are trying to say.

**Response: We have removed this section to improve clarity.**

Lines 141 through 145. I really don't understand the relevance of this statement of some other results of other study that has been placed here in the methods section. This seems like it could be eliminated.

**Response: We have removed this section to improve clarity.**

Lines 189 through 192. Again, the way this is written, it would seem that this samples described in this current submission or actually collected and described previously in the Benson study. This needs to be made clear whether it is true or not.

**Response: This sentence has been revised to improve clarity.**

Lines 196 through 199. I am a bit surprised that they made boats out of parafilm itself rather than using paraffin coated aluminum that has been more typically used in such droplet freezing assays. I would have expected that the paraffin would act as an insulator and that the droplets may not be as close to the temperature of the ethylene glycol coolant as they would have been on an aluminum surface. This may not be fatal, but did surprise me.

**Response: The cryoboat method for testing for ice nucleation activity using parafilm has been used extensively in previous publications. We have added references to showcase this method.**

Lines 201 through 205. The way this method is described, it would seem to suggest that they froze only two droplets. I would hope this is not correct. They definitely need to make this clearer. If in fact they only drop froze 2 droplets I would be very disappointed, given that there is a decent possibility of spontaneous freezing of some of

the droplets, and this might have given them false positive responses. While the frequency of nucleation in control droplets that would not contain ice+ bacteria is quite low, with such small numbers of droplets, they might well have misled themselves in some cases if only two drops were tested.

**Response: Succinct details have been added on the droplet freezing assays and the number of repetitions. These procedures have been followed as per previous publications (e.g., Garcia et al., 2019; Pietsch et al., 2016).**

Lines 225 to 227. It is unclear what they mean by "putative P. syringae". Did they previously think they were P. syringae and then ended up grouping with Pseudomonas fluorescence.? To refer to the "83 branched with the reference strain" is very awkward. Certainly "branch" is not the right word – "grouped is better.

**Response: We have made significant changes in terminology in several sections, and have added a new supplementary figure to highlight the selections and classification scheme.**

Line 256. I really don't know what they mean by "compositional resemblance".

**Response: This has been changed to reflect richness based on genera.**

Lines 290 to 292. The sentence is very awkwardly written. Instead of saying that phylogroup 02 "comprised of collections" from various habitats, it would be much more accurate to say that phylogroup 02 "include strains that have been collected from these various habitats". The same issue is also present in their description of the other phylogroups.

**Response: This sentence has been changed as suggested.**

Lines 301 through 315. I found this long discussion comparing their results to that of a previous study done in Virginia to be very awkward and unnecessary. I didn't see that it added much - and it could have been summarized in a single sentence saying that a similar frequency of ice+ P. syringae was observed as in virginia.

**Response: This has been deleted from the discussion as suggested. A single sentence summarizing that study was moved to the introduction.**

Line 322 through 324. I don't understand the logic in this sentence. Furthermore, towards the end of the discussion they have extensively referred to the likelihood that the surrounding vegetation and land use etc was contributing to the differences in the numbers and types of bacteria seen in these lakes. This seems very likely, but at the same time, readers were not given any information that would help us get a better feel for whether this is likely. That is, a table or some other summary of the features of the lakes and the surrounding area would definitely have helped readers make more conclusions about how to interpret the results of this study. I would strongly urge additional information to be provided about the surrounding area and hence the meta community that probably led to the microbes seen in the water.

**Response: This section has been removed to avoid confusion. The paragraph has also been re-structured to reference our hypothesis.**

Line 330 through 331. This very awkward sentence is very hard to understand.

**Response: This sentence has been deleted.**

Lines 332-335. Again, a more detailed description of the lakes in the surrounding region would be very helpful here.

**Response: We have added a few more descriptions.**

Lines 336 through 345. This material all belongs in the results, and not in the discussion, since there is no discussion of it here.

**Response: This paragraph has been re-structured, with large portions moved to the results as suggested.**

Line 347-348. This sentence is very awkward. I don't understand it.

**Response: The sentence has been deleted.**

Line 358 through 359. I am still having trouble understanding what they refer to as a " hotspot". Has such localized

concentration of bacteria in open waters been described by others? In a freely mixed body of water like a lake, it is hard to imagine how it could occur. Also, I don't quite understand how the whole vectoring story applies only to a hotspot. I would assume that the processes such as wind splash etc that would release bacteria from a lake would occur equally likely at any location on the surface of the lake, and not only at the hotspot.

Response: This paragraph has been re-structured. We have added additional language to define a hot spot, and also a reference to the relevance of these high concentration areas to aerosolization processes.

Lines 363 through 367. I don't understand any of this. I don't quite see how looking at a surface of water is going to give us much information about what was going on within the water body itself.

Response: This paragraph has been deleted. We moved one sentence up with relevance to surface scum associated with harmful algal blooms, representative of hot spots of cyanobacteria on the surface of lakes.

Lines 369 through 384. The conclusions section seems to simply repeat everything we've already read earlier. It can easily be dropped.

Response: The conclusions section has been deleted.

Reviewer #2

## Basic reporting

The manuscript presents data associated with sampling via drones of lakes at different elevations in Austria to identify what is referred to as Bona fide Pseudomonas syringae populations. Although the introduction describes previous literature relating to sampling from noncrop locations previously, there is a need for basic information on Pseudomonas syringae leading up to the work. For instance phylogroup is used but what is the relevance of phylogroup in this manuscript. Secondly, the authors use bona fide to describe P. syringae. The authors should also introduce sensu lato and stricto sensu given they are using P. syringae in a very broad sense.

Response: Thank you for thorough review. We have provided additional background regarding *P. syringae* and phylogroup designations. We have added additional information and references about sensu lato and sensu stricto in reference to *Pseudomonas syringae* and have removed bona fide throughout to improve clarity.

## Experimental design

The experimental design seems to be fine except that it would help to provide information on the method of sampling via drones.

Response: We have provided additional details on the drone-sampling methods throughout the manuscript.

## Validity of the findings

It is difficult to assess some of the findings as they are not presented as best I can see. Thirty-two haplotypes were identified in the manuscript although these are not obvious in the manuscript. Why not include a supplementary file so reader can judge. In the results in figure 2 four phylogroups are shown although in results eight pgs are indicated. Why not include these in a supplementary table? Why is there not a Shannon's diversity index to make the points as to relative diversity. Other lakes also had a high number of genera. Rather than focusing on the diversity associated with elevation, it would be more interesting to focus more on the unique bacterial populations in three lakes indicated in the NMDS analysis.

Response: We have included supplementary data on the haplotypes as suggested. Of the 32 haplotypes observed more than once in any lake, only 11 were observed in at least two lakes. These mapped to only eight phylogroups. The remaining unique haplotypes are included as black bars in Figure 2. The legend has been updated to improve clarity. We have updated Fig. 2 to include HP32; now there are 11 HT in the figure representing those observed in at least two lakes.

Note: The reviewer has attached an annotated manuscript to this review.

Response: Thank you for for the annotated manuscripts revisions. We have addressed all of your edits and highlighted revisions:

- rDNA added in abstract
- bona fide removed, sensu stricto described, terminology adjusted throughout
- drone-sampling methods added to introduction
- NIST SI units were used for mL (instead of ml), which appears appropriate for PeerJ
- Line 135-138. This section has been deleted.
- Line 150. 5 min. template boil
- L245-246, language updated with new terminology and clarified for Table 2.

Basic reporting

The manuscript from Hanlon et al. describes the isolation and characterization of water samples from a variety of Austrian lakes. The authors sample water, and plate out on either TSA or KBC media to isolate/identify total cultureable bacteria and potential Pseudomonas syringae strains (respectively). The authors also report on metrics and statistics concerning cultureable bacterial communities within the lakes and report on ice nucleation potential of Pseudomonads. There is an interesting relationship reported between elevation of the lake and ice nucleation percentage of the strains.

Overall, I felt that the data collection and analyses were adequate for publication. However, I also felt that the manuscript could be significantly streamlined and reconfigured to be more clear about the questions being tested and how the experiments addressed these questions.

**Response: Thank you for your careful review of the manuscript. As noted in our response to Reviewer 1, we have overhauled the manuscript.**

Abstract: The abstract is far too long. Please condense and streamline this so that it reads as one/two paragraphs and does not include fine scale data and interpretations of the data. Please reshape the abstract so that it reflects quick take home messages of the manuscript.

**Response: The abstract has been reduced significantly.**

Introduction/Discussion: I think these sections could be reconfigured and reshaped as well, to better reflect the questions being asked and how the experiments address the questions. The authors move back and forth between analyses of cultureable bacteria from the lakes to Pseudomonad percentages to ice nucleation percentages, but the intro and discussion as is reads as a bit too scattered and could be made much more linear and direct. The authors should also highlight why they are investigating community diversity in the context of Pseudomonads and what these analyses actually tell the reader. I went in thinking that this paper would be concise and report back on number of ice nucleating strains per lake (which it ultimately does, just in a roundabout way). Please include additional context and transition sentences and information that helps the reader to understand why the additional community level analyses and statistics are being reported.

**Response: The abstract has been reduced significantly. The introduction has been re-organized and re-focused. The discussion has been significantly reduced to focus on the main points, and additional references have been added where appropriate.**

Experimental design

I think the methods and analyses are quite clear, although the authors should include the number of times that the ice nucleation tests were performed (they should have at least been repeated). Please include information on replication of the ice nucleation assays prior to resubmission.

**Response. Additional information has been added regarding the ice nucleation assays. This concern was also raised by another reviewer.**

Validity of the findings

I think the results are pretty straightforward and reported as such. No other comments.

Additional comments

As I mentioned above, I think the authors should consider how to reconfigure the intro and discussion so that the take away message of the manuscript is clear and concise. I have no problem with descriptive papers, I just want to be able to capture the take home message after reading.

**Response: We trust that the overhauled manuscript has addressed these concerns.**